



# Characterisation of subglacial water using a constrained transdimensional Bayesian Time Domain Electromagnetic Inversion

Siobhan F. Killingbeck[1], Adam D. Booth[1], Philip W. Livermore[1], Charles R. Bates[2], Landis J. West[1]

[1]School of Earth and Environment, University of Leeds, Leeds, LS2 9JT, UK
[2]Earth and Environmental Sciences, University of St Andrews, St Andrews, KY16 9AL, UK

*Correspondence to*: Siobhan F. Killingbeck (eespr@leeds.ac.uk)

**Abstract.**

Subglacial water influences the dynamics of ice masses. The state of subglacial pore water, whether liquid or frozen, is
associated with differences in electrical resistivity that span several orders of magnitude, hence liquid water can be inferred
from electrical resistivity depth profiles. Such profiles can be obtained from inversions of time domain electromagnetics (TEM)
soundings, but these are often non-unique. Here, we adapt an existing Bayesian transdimensional algorithm ('MuLTI') to the
inversion of TEM data constrained by independent depth constraints, to provide statistical properties and uncertainty analysis
of the resistivity profile with depth. The method was applied to ground-based TEM data acquired on the terminus of the
Norwegian glacier Midtdalsbreen, with depth constraints provided by co-located ground penetrating radar data.  Our inversion
shows that the glacier bed is directly underlain by material of resistivity $10^2$ Ωm ±100 %, with thickness 5-40 m, in turn
underlain by a highly conductive basement ($10^0$ Ωm ±15 %). High resistivity material, $5\times10^4$ Ωm ±25 %, exists at the front of
the glacier. All uncertainties are defined by the interquartile range of the posterior resistivity distribution. Combining these
resistivity profiles with co-located seismic shear-wave velocity inversions to further reduce ambiguity in the hydro-geological
interpretation of the subsurface, we propose a new 3D interpretation of the Midtdalsbreen subglacial material partitioned into
partially frozen sediment, frozen sediment/permafrost and weathered/fractured bedrock with saline water.

## 1 Introduction

Subglacial structure and material properties are one of several important controls on ice flow, both through composition and
ice/material interactions. The potential for subglacial sediments to store and pressurise water is a key element in predicting the
evolution of ice masses of all sizes, from small mountain glaciers to large polar ice-sheets (Christoffersen et al., 2014; Siegert
et al, 2018). Currently, the ability to develop accurate ice flow models is limited by poor understanding of processes acting at
the ice/bed interface, and the composition of subglacial material. With increased knowledge of subglacial structure and
sediment liquid water content, our ability to predict glacier retreat patterns is greatly increased.



Non-invasive geophysical imaging methods are widely and successfully applied to characterise the internal properties of glacier ice and its immediate basal environment. Such methods (including reflection seismology and ground-penetrating radar) can underperform when characterising material properties beyond the first few metres of the glacier bed (Booth et al., 2012), yet subglacial aquifers, sediment accumulations and permafrost can extend to much greater depths (e.g., Mikucki et al., 2015 and Hauck et al., 2001). Further still, inversions of isolated geophysical datasets are unconstrained and non-unique, with many models of the subsurface matching the observed dataset. Joint inversions using multiple independent datasets can constrain the model space, combining depth and resolution sensitivities from multiple datasets. In this paper, time-domain electromagnetics (TEM) and depth constraints derived from ground penetrating radar (GPR) are combined in a constrained inversion to provide geophysical insight into the structure and water characteristics of the subglacial environment. This method is adapted from a transdimensional Bayesian framework, termed 'MuLTI' and described in Killingbeck et al. (2018), originally applied to characterise subglacial sediment distribution from seismic surface wave data (Killingbeck et al., 2019). Here, we explore a similar concept for the TEM method.

TEM methods use electromagnetic fields to investigate subsurface resistivity structure by measuring eddy currents induced by an offset transmitter coil. The method has a depth sensitivity ranging from a few meters to kilometres, depending on the survey parameters used. Of particular relevance here is that electrical resistivity increases by several orders of magnitude when water in pores freezes (Hoekstra and McNeill, 1973), allowing resistivity methods to indicate the liquid water content of subsurface materials. TEM methods have been extensively applied for hydrogeophysical exploration to map groundwater resources (Auken et al., 2003), mapping permafrost on mountainous regions under debris covered glaciers (Hauck et al., 2001), mapping arctic permafrost in Alaska (Minsley et al., 2012), and more recently mapping deep saline groundwater zones in Antarctica's Taylor Valley (Mikucki et al., 2015). These studies illustrate that characterising the resistivity of the subsurface offers a promising means of distinguishing material type and water content within the subglacial environment.

In TEM, the resistivity-depth profile is not directly measured but rather inferred from the measured eddy currents. In common with most geophysical inversions, this profile is non-unique, implying that many profiles fit the data within error tolerance, and averaging is usually employed to recover a single solution. Early inversion techniques for TEM data included non-linear least squares (Barnett, 1984) and an Occam-type regularization method to obtain a smooth solution (Constable et al., 1987), but these were prone to being trapped in local minima with any large resistivity variations becoming smoothed. More recent inversion methods include laterally- and spatially-constrained algorithms to regularize the inversion and obtain solutions that agree with the expected geological variations (e.g., Christensen and Tølbøll, 2009; Vignoli et al., 2015; Auken et al., 2015). Yet, these methods do not provide detailed uncertainty analysis of the estimated model parameters and require a fixed number of layers in the model. The maximum depth of investigation (DOI) is generally estimated using methods, such as half-space skin depth (Spies, 1989) or the Jacobian sensitivity matrix (Christiansen and Auken, 2012), though these do not consider the non-linear sensitivity of the DOI to conductivity structure. These limitations in uncertainty quantification, fixed model space and DOI estimation can be mitigated by transdimensional Bayesian sampling-based inverse methods. These produce an ensemble of models from which statistical properties of the model parameters, including model dimensions, can





be inferred (Mosegaard and Tarantola, 1995, Blatter et al., 2018). The computed posterior probability density function (pdf) provides a robust measure of DOI, highlighting model uncertainty at each depth (Blatter et al., 2018). To further reduce the parameter space and improve vertical resolution, the inversion can be constrained with complementary depth information (e.g., from borehole records or other geophysical sources), provided that the constrained depth is consistent across datasets.

5        In this paper, we derive the implementation of 'MuLTI-TEM' (Multimodal Layered Transdimensional Inversion of Time Domain Electromagnetics) and its use for characterising the subglacial environment. After testing the method on a synthetic dataset, we analyse a TEM dataset acquired on the Norwegian glacier Midtdalsbreen, an outlet of the Hardangerjøkulen ice cap, using complementary GPR data for constraining the ice thickness. Since the glacier bed represents a transition in both dielectric constant and electrical resistivity, the GPR depth constraint can be used directly in the TEM

inversion. Recent results from Killingbeck et al. (2019), interpret the Midtdalsbreen subsurface, from seismic shear wave velocity (Vs) profiles, as local accumulations of soft material (partially frozen glacial till) and permafrost overlying bedrock. Finally, we show a combined 3D interpretation of previous shear wave inversions (Killingbeck et al., 2019) and results output from MuLTI-TEM, and suggest the development of a joint resistivity-Vs depth-constrained inversion strategy.

## 2 Method

### 2.1 Time Domain Electromagnetics

In TEM surveying, an electromagnetic field is generated by sending a periodic, modified square-wave, current through a transmitter coil. When the current is on, a static electromagnetic field is established in the ground. The electromagnetic field is varied by terminating the current abruptly at the first quarter-period, being reduced to zero for the second quarter-period, the current is then reversed for the third quarter-period before being reduced to zero again for the final quarter-period. This time-

dependence induces eddy currents in the subsurface, initiating within the immediate vicinity of the transmitter then spreading downwards. The eddy currents induce a secondary electromagnetic field which propagates back up through the subsurface, inducing a current in a receiver coil located at some distance from the transmitter. The receiver measures the induced secondary electromagnetic field in the transmitter-off periods. The response of the subsurface is measured in terms of the decaying amplitude of the secondary electromagnetic field. This is recorded as a function of time, with later responses originating from

greater depths. With regards to conductivity of the subsurface, the more conductive the subsurface, the larger the eddy currents and the larger the measured secondary electromagnetic field will be. By taking repeated measurements a sounding curve, similar to DC resistivity soundings, is obtained (Geonics, 1994). The measured voltages versus time from the receiver coil are then used to constrain the resistivity profile with depth.

       The maximum depth of investigation (DOI) ($h$), in meters, is:

$h \approx 8.94 \, l^{0.4} \rho^{0.25}$ ,                                                     (1)




where $l$ is the transmitter loop size in meters and $\rho$ is the upper layer specific resistivity in Ωm (Geonics, 1994). Equation (1) shows the transmitter loop size is an important acquisition parameter controlling depth of investigation. Large loop soundings (e.g., > 40x40 m), where the receiver coil is located in the centre of the large transmitter coil, have been conducted on thick permafrost regions by Rozenberg and others (1985) and Todd and Dallimore (1998). Our depth of target (0 - 80 m) allows for a smaller, more portable loop size to be used. In this study, we use an efficient 10 x 10 m loop, with the receiver 15 m offset from the centre of the transmitter loop (to stop electromagnetic interference with the receiver) shown and discussed further in section 3.

### 2.2 MuLTI-TEM

MuLTI-TEM is a Bayesian inversion Matlab code that determines the posterior distribution of resistivity as a function of depth. It is adapted from the MuLTI algorithm ('Multimodal Layered Transdimensional Inversion), developed for seismic surface wave inversions (see detailed in Killingbeck et al. (2018)). The data input, $d$, to MuLTI-TEM are the measured voltages ($v$) at each timegate ($t$), together with an estimate of their uncertainty ($\sigma$) derived from the variance of each data point calculated from the stack recordings:

$$d = [t_1, t_2 \ldots. t_N, v_1, v_2 \ldots. v_N], \tag{2}$$

$$\sigma = [\sigma_1, \sigma_2 \ldots. \sigma_N], \tag{}$$

The method used to find the posterior distribution of the resistivity profile is outlined below,

$$p(m|d) = p(d|m)p(m)/p(d) \tag{3}$$

where $p(m|d)$ is the posterior probability of the model ($m$) given $d$, $p(m)$ is the prior information known about the model, $p(d|m)$ is the likelihood and $p(d)$ is the evidence. A Markov Chain Monte Carlo methodology is used to sample the posterior distribution, traversing the space of admissible models with the statistics of the ensemble converging to the underlying posterior distribution, provided the chain of models is long enough.

We describe the 1D variation of resistivity with depth as a piecewise constant function using Voronoi nuclei (see Killingbeck et al., 2018). Any available depth constraints separate the resistivity into different depth layers. In our case, in which constraints are drawn from GPR data, we consider depth constraints to be exact since the accuracy of GPR depth estimation is ~100-times smaller than the thinnest resolvable layer in TEM. Within each layer, we define a single confined nucleus; aside from being confined to the given layer, this nucleus is otherwise unconstrained in depth. The number of confined layers, $l$, is equal to the number of layered depth constraints applied with the addition of an assumed half space of constant resistivity extending to infinite depth. If no depth constraints are applied then $l = 1$, corresponding to a half space. We add in an additional $k$ nuclei in the model that are unconstrained in depth, termed *floating*. Our transdimensional framework allows the data to self-determine the required number of layers $k$ (e.g., Bodin and Sambridge, 2009; Bodin et al., 2012, Livermore et al., 2018), thus $k$ is also an unknown that we determine.

The model vector, that describes the resistivity profile, is then

$$m = [dp_1, dp_2 \ldots. dp_k, \log(R_1), \log(R_2) \ldots. \log(R_k), k, dpc_1, dpc_2 \ldots. dpc_l, \log(Rc_1), \log(Rc_2) \ldots. \log(Rc_l)], \tag{4}$$





where $dp_i$ are the floating nuclei depths, $\log(Rc_i)$ are the base-e log of the their respective resistivities, $dpc_i$ are the confined nuclei depths and $\log(Rc_i)$ are the base-e log of their respective resistivities. In our transdimensional framework the number of floating nuclei ($k$) is a free parameter and self-determined in the algorithm.

The prior distribution on the model parameters depend on which layer each nuclei is within. If no depth constraints of the subsurface interfaces are available (that is, there is only a single layer of the half space), the prior distributions on the resistivity is uniform with wide bounds on log(R) (e.g., R between $10^0$-$10^5$ Ωm), as no prior information (beyond that which can be reasonably assumed for typical materials) is known about the subsurface. However, when depth constraints also provide lithological information, the range of $R$ can be tightened, thus significantly reducing the model parameter space.

Lastly, the likelihood is defined by assuming that the measurements are normally distributed about values calculated from a forward model of TEM response (assuming a given resistivity profile) and the estimated standard deviation $\sigma$. MuLTI-TEM uses the Leroi algorithm of the CSIRO and AMIRA project P223F (CSIRO and AMIRA, 2019) as a forward modeller to compare proposed subsurface models to the observed data. The Leroi algorithm is written in Fortran 95 and has a wide range of electromagnetic modelling capabilities, for more information see Raiche (2008). We have used a simplified version of this algorithm and created a mex file to call the code in Matlab from within the MuLTI-TEM algorithm. See Fig. A1 in the appendix for the detailed Leroi input file on which our simplified code is based.

MuLTI-TEM numerically approximates the posterior distribution by creating an ensemble of models, traversing the model space and sampling the models with greater likelihood more often than models with a poor fit to the observed data. Provided the ensemble, is sufficiently sampled the numerically-obtained posterior distribution will converge to the true posterior. This is achieved by constructing a Markov-chain, each model in the chain being based on the previous model but randomly perturbed, the size of the perturbation being controlled by the user. MuLTI-TEM produces a variety of statistics of the resistivity ensemble including, but not limited to: the mean and mode (the most likely) solution and 95% credible intervals as an estimate of its uncertainty, thus giving a profile with a quantified uncertainty.

## 3 Data Acquisition

Data acquisition was performed on Midtdalsbreen, a NE-flowing outlet glacier of the Hardangerjøkulen ice cap in central-southern Norway (60.59ºN, 7.52ºE; Fig. 1a) in April-May 2018. Midtdalsbreen is surrounded by mountains of phyllite, crystalline granite and gneiss suggesting this as the underlying bedrock. The glacier is well-suited to methodological development as it is logistically accessible, especially with multiple types of geophysical surveying equipment.

GPR, seismic and TEM surveys were performed around and over the glacier front (Fig. 1). All methods were acquired at each line highlighted, A-D, in the same field season. Lines B and C are located entirely on the glacier, whereas Line A shows no glacier ice. Line D traverses through each of Lines A, B and C and extends beyond the glacier terminus. At the time of acquisition, the subsurface comprised snow (2-4 m thick) overlying a varying thickness (0-25 m) of glacier ice, and a

 

substrate of unknown subglacial material. This layered interpretation is based on the interpretation of the GPR dataset, which also suggest that the snow and ice layers show little variation in any of lines A, B and C.

Killingbeck et al. (2019) used MuLTI to jointly interpret the seismic and GPR data, defining regions of partially frozen sediment and hard bedrock based on subglacial shear wave velocities (Vs < 1000 m/s for the former, > 2000 m/s for the latter. Fig. 1b shows the seismic Vs results obtained at the glacier bed along all survey lines. This paper focuses initially on the joint inversion of the TEM and GPR data, and thereafter integrates the observations with the existing Vs distributions.

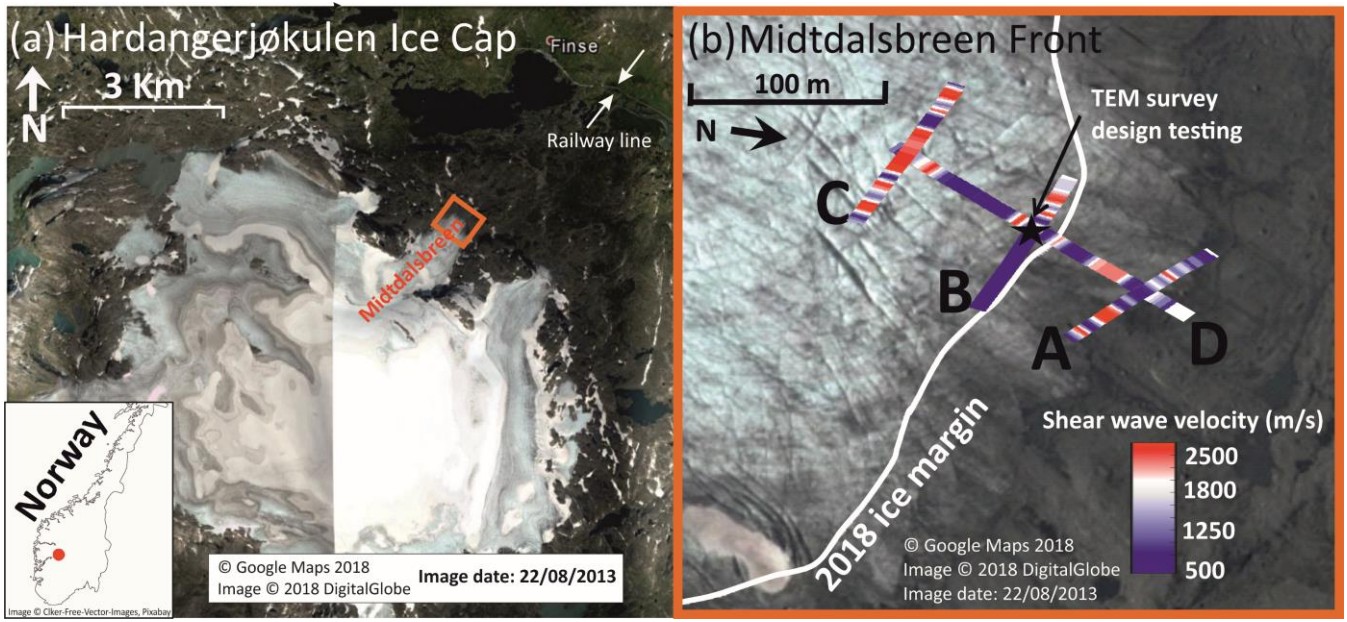

**Figure 1. a) Location of Hardangerjølken ice cap in Norway and Google Earth image of Midtdalsbreen, survey area, with nearest sources of TEM noise (town and railway line) highlighted. b) Survey lines acquired during 2018 field season with the seismic Vs results obtained at the top rock horizon displayed. The orange border around (b) identifies the same area as the orange box in (a), note (b) is rotated away from North to enable optimal data comparison in later figures.**

TEM data were acquired with a Geonics PROTEM 47 system, consisting of a 3 channel digital time-domain receiver, a TEM-47 battery powered transmitter and a 3D multi-turn receiver coil. All survey parameter are listed in Table 1. For cross-glacier lines A, B and C, the system was moved along the lines in 4 m intervals; for the longer down-glacier line D, this was increased to 8 m. Multiple survey configurations were initially tested at the intersection of lines B and D to determine the optimal survey configuration for imaging the subglacial environment at Midtdalsbreen. These tests comprised the following (the maximum DOI of each test is estimated using Eq. (1) and upper layer (snow) resistivity as 30 to 100 Ωm):

a)  37 m x 37 m square transmitter with receiver in centre, with estimated DOI 90 to 120 m (Fig. 2a)

b)  10 m x 10 m square transmitter with receiver 15m away from centre of transmitter square, with estimated DOI 50 to 70 m (Fig. 2b.)

c)  5 m x 5 m square transmitter with receiver 12.5m away from centre of transmitter square, with estimated DOI 40 to 54 m (Fig. 2c.)



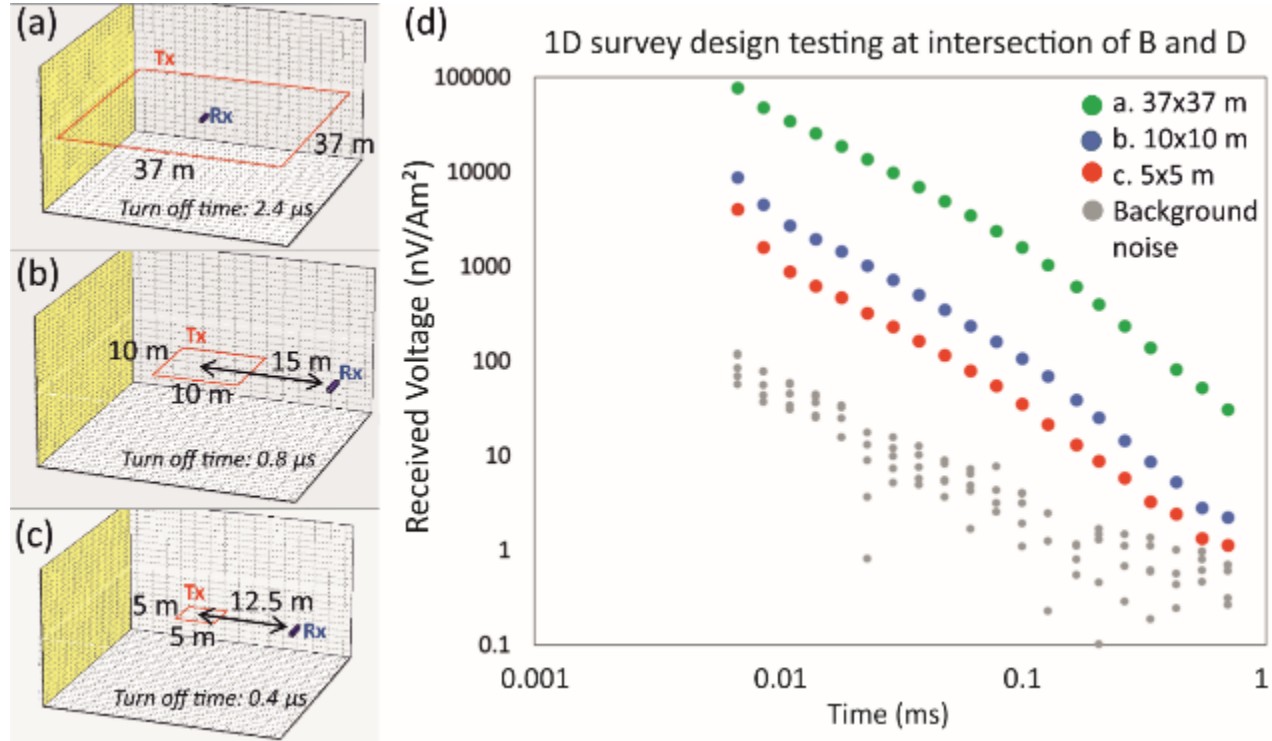

**Figure 2. Survey configuration testing at the intersection of B and D. a) 37m x 37m transmitter coil with receiver in the centre. b) 10m x 10m transmitter coil with receiver 15m offset. c) 5m x 5m transmitter coil with receiver 12.5m offset. d) Raw data acquired at the intersection of B and D (237.5Hz), from each survey configuration, plotted with background noise recorded with transmitter turned off.**

For accurate resistivity soundings, the raw signal should be above background noise levels (Fig.2d). Background noise, measured with the transmitter coil turned off, is considered low at Midtdalsbreen since there are no large sources of electrical noise e.g. power lines, buildings, roads, metal infrastructure for ~5 km (where the nearest town, Finse, is located, Fig. 1). The 10 m x 10 m square transmitter (Fig. 2b, Fig. 3) was chosen as the optimum survey configuration because:

    i)        it had a fast turn off time (for imaging the shallow surface),

    ii)       the raw signal (received voltage) recorded was sufficiently greater than the background noise (Fig. 2d),

    iii)     it was easily deployed on the glacier, and could be moved rapidly between the points of the survey lines (Fig. 3), and

    iv)     the maximum DOI, 50-70 m, is sufficient for imaging our target depth, subglacial sediments below ~25 m thick ice.





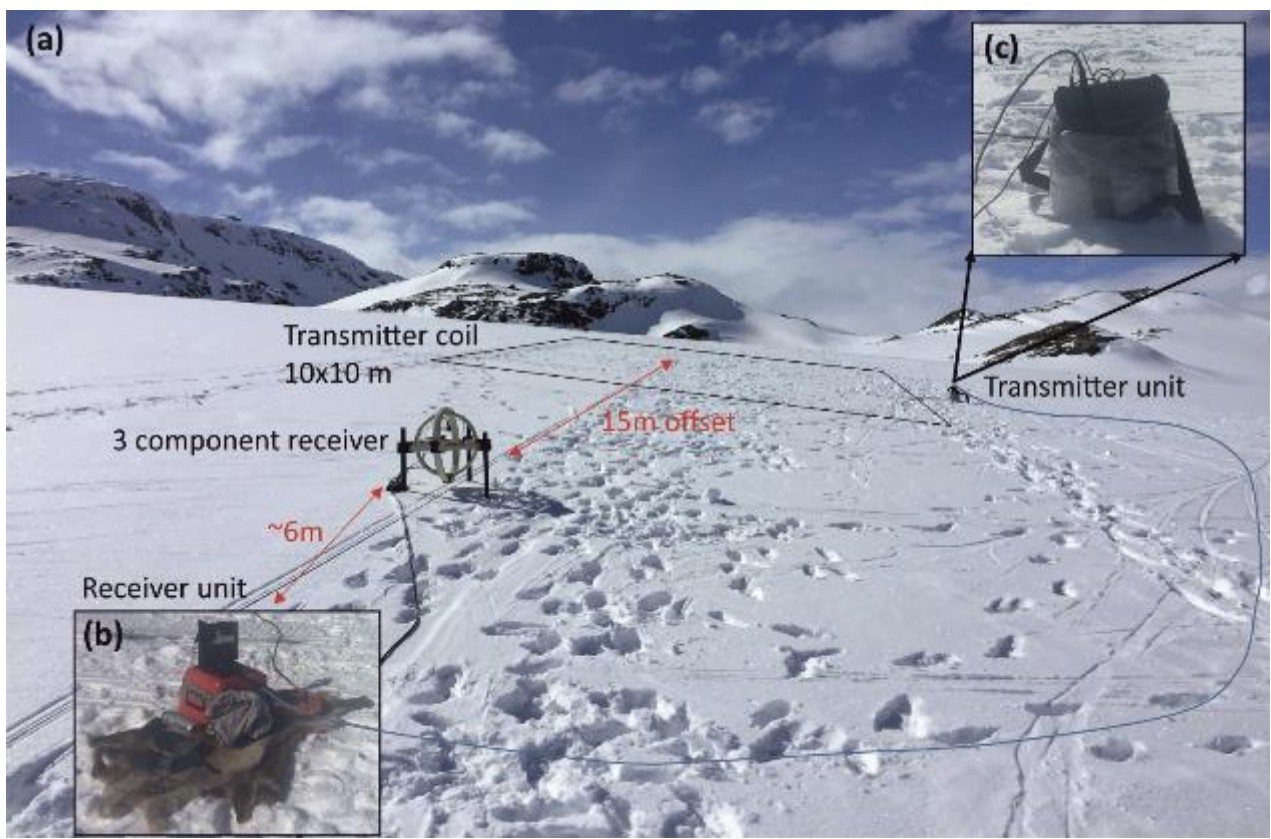

**Figure 3. a) Survey configuration used for acquiring lines A, B, C and D on the glacier. b) Image of the receiver unit on top of a rug to protect unit from snow and easily drag along the lines. c) Image of transmitter unit sitting in bubble-wrap pocket used to protect unit and batteries from snow and cold.**





**Table 1. TEM survey parameters**

| Fixed Survey Parameters | |
|---|---|
| Transmitter coil | 10 m x 10 m |
| Transmitter – receiver offset | 15 m |
| Turn off time | 0.8 us |
| Gain | 4 |
| Frequency | 237.5 Hz |
| Orientation of Tx and Rx | 220 degrees |
| Gate | 20 |
| Integration Time | 15 seconds |
| Repetition Base | 50 Hz |
| Receiver effective area | 31.4 m$^2$ |
| Number of repeat readings stacked | 3 |
| Transmitter current | 2 A |

## 4 Application of MuLTI-TEM to a synthetic dataset

Synthetic TEM responses from a variety of models representing different possible glacial and subglacial structures of the
5   Midtdalsbreen glacier were input into MuLTI-TEM for validation (models (a)-(e) in Fig.4). Each model included layers of
snow, ice, and bedrock, with models (b)-(e) also including saturated subglacial sediment. Each layer was populated with
representative resistivities from previous TEM studies (Mikucki et al., 2015). Certain models were designed to test particular
aspects of the inversion: model (b) tested the maximum DOI using our specified survey design and synthetic, and model (c)
tested whether the inversion can resolve a 5 m-thin layer.

10        Synthetic TEM responses were calculated from the 1D block models using the Leroi forward modelling algorithm
(Raiche, 2008), then 5% normally distributed random noise was applied to all time gates, a similar noise model to Blatter et
al. (2018), see Fig. 5. The simulated TEM survey configuration assumed a 10 m x 10 m square transmitter with receiver 15m
away, consistent with the field acquisition.





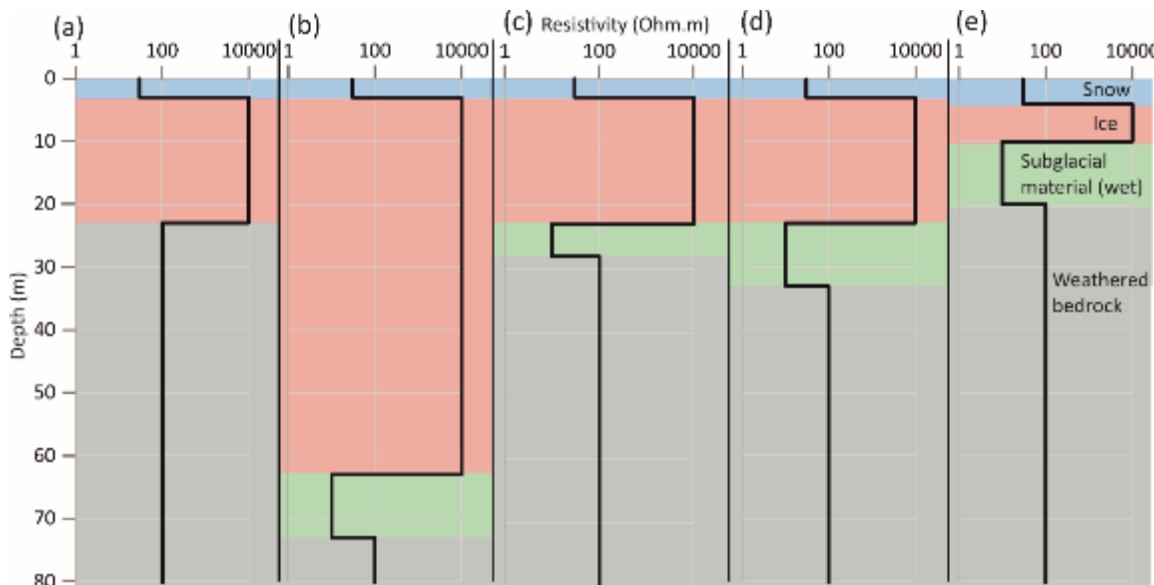

**Figure 4. 1D synthetic block models created to simulate different subsurface scenarios expected at Midtdalsbreen.**

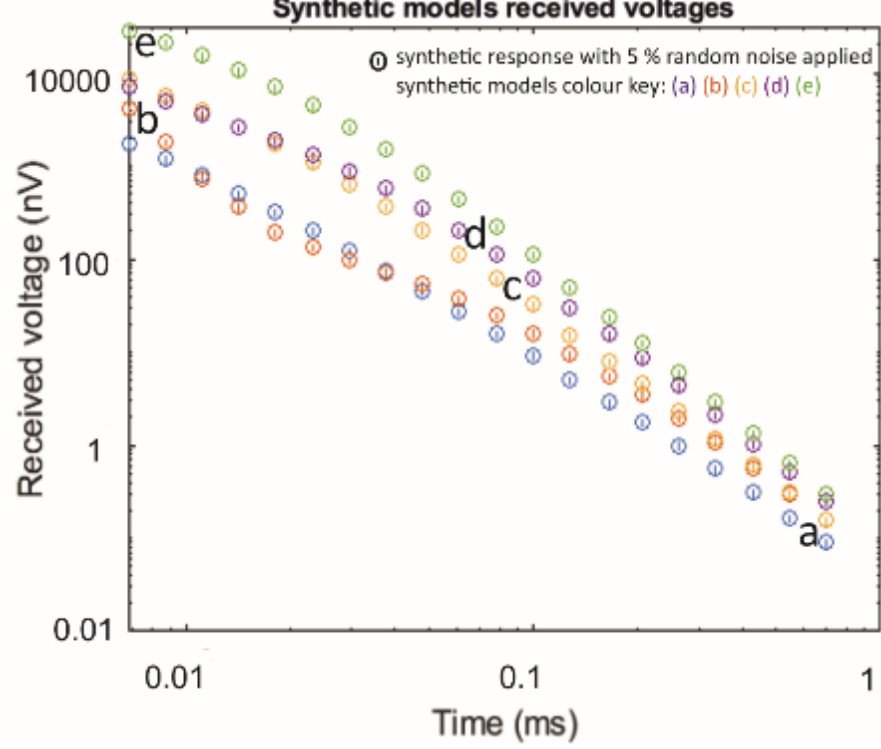

**Figure 5. Forward modelled responses for 1D synthetic block models a-e with 5 % random noise applied. The lines within the circles represent the 5% error bars.**





The inversions were run using resistivity ranges shown in Table 2 using a maximum depth of 80 m, consistent with the estimated maximum DOI for this configuration. To highlight the benefit of additional depth constraints, MuLTI-TEM is separately run for an unconstrained case, and a case in which the depths of snow-ice and ice-bed horizons are fixed. One million iterations were sufficient for the posterior distribution to converge (a test of 2 million iterations produced the same

posterior). Additional figures in A2 show that the modal model from the depth constrained inversion fits the data better than that from the non-constrained inversion, with, in general, a lower data misfit. Multiple chains were also tested with one million iterations using different initial conditions. For the constrained case, these produced similar posterior distributions with identical interpretation, indicating that only one chain was needed. For the unconstrained case, the posterior distributions differ slightly but are nevertheless qualitatively the same, suggesting that the unconstrained case is not yet converged (Fig. A3).

More detailed inversion parameters used are documented in Table A1.

**Table 2. Resistivity parameter boundaries used in MuLTI-TEM for the glacier feasibility study.**

| Material | Resistivity boundaries ($\Omega$m) |
|---|---|
| Snow | $10^0$-$10^3$ |
| Ice | $10^3$ - $10^5$ |
| Subglacial material | $10^0$ - $10^5$ |
| Non constrained material | $10^0$ - $10^5$ |

Posterior probability density distributions (pdf) of the synthetic resistivity profiles produced from MuLTI-TEM are shown in Fig. 6. These are shown, within their 95% credible interval, as coloured contours, where red indicates the most likely

values. Consistent with many previous studies, both the unconstrained and constrained inversions indicate that the TEM method can resolve conductive structures much more accurately than resistive ones, highlighted by the much tighter pdf over the conductive sediment layer compared to the resistive ice layer. The unconstrained inversions (Fig. 6a) capture a similar structure to the true model, but they struggle to resolve true layer depths. The simple synthetic model (a) and thick resistive layered model (b) are relatively well resolved, however the more complicated synthetic models with thin layers and large

resistivity contrasts (c, d and e) are not. The depth-averaged resistivity errors within the subglacial layer (calculated from the difference of the modal and true solutions) of each synthetic non-constrained solution are: a) 275 $\Omega$m, b) 14 $\Omega$m, c) 30 $\Omega$m, d) 21 $\Omega$m and e) 2000 $\Omega$m. The addition of depth constraints (Fig.6b) improves the match throughout. The resistivity of the thin snow layers and conductive sediment layers are well-resolved in all synthetics, with depth-averaged resistivity errors within the subglacial layer reduced to: a) 8 $\Omega$m, b) 14 $\Omega$m, c) 23 $\Omega$m, d) 10 $\Omega$m and e) 25 $\Omega$m (a factor of 34, 1, 1.3, 2.1 and 8

improvement on their unconstrained equivalents). Note, imaging beneath a conductive structure is difficult for the TEM method due to the attenuated signal (Blatter et al., 2018); however our constrained inversion results show the bottom of the conductive layers (in (b) – (e)) to be much better resolved, i.e. to within < 10 meters.





The TEM method is generally more sensitive to conductance (the product of conductivity and thickness) rather than the layer conductivity or thickness alone (Geonics, 1994). Therefore, modelling is challenged by a non-unique problem, for example with thinner more conductive layers producing a similar TEM signal to thicker less conductive layers. The addition of depth constraints greatly reduces this non-uniqueness enabling more accurate solutions to be obtained at all depths.

5    However, an example where this TEM inversion struggles is when a thin conductive layer exists above a resistive basement (Fig.A5). In this example (Fig. A5 left panel), a thin layer 1 m thick and resistivity 1 $\Omega$m has conductance of 1 S, equivalent to a 10 m thick semi-conductive layer of 10 $\Omega$m resistivity. Inverting this synthetic example with depth constraints in MuLTI-TEM shows that such a thin conductive layer above resistive basement cannot be fully resolved even by the constrained inversion, appearing as a much thicker, less conductive layer.

10    This feasibility study highlights the significant added value of depth constraints when a complex resistivity structure is expected. It demonstrates MuLTI-TEM is promising for the potential distributions of resistivity beneath Midtdalsbreen.

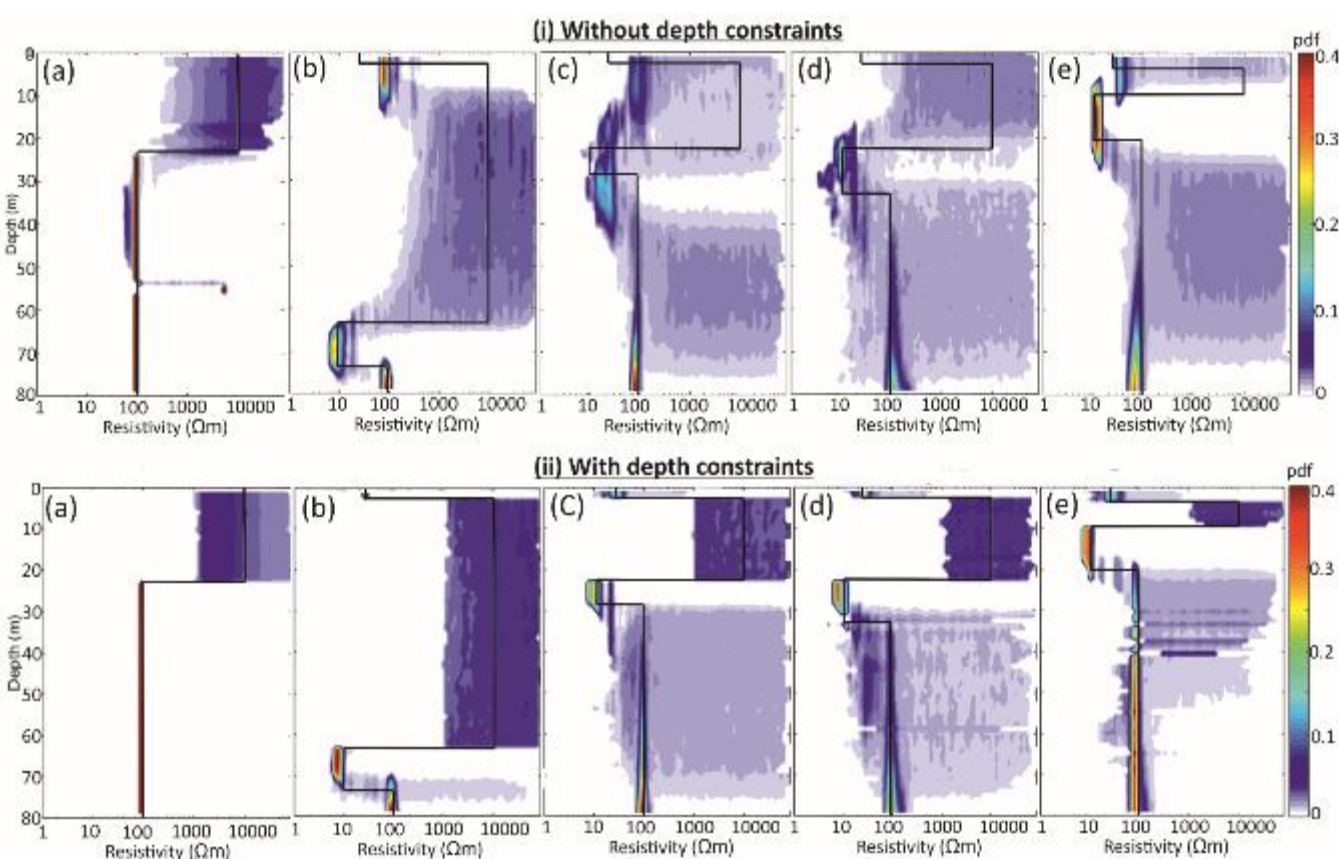

**Figure 6. Posterior distributions of resistivity determined from MuLTI-TEM inversion. Top: i) without depth constraints. Bottom: ii) with depth constraints. The models correspond to those shown in Fig. 4a-e, highlighted by the black line. The colour scale represents the probability density distribution of resistivity within the 95% credible interval.**





## 5 Application of MuLTI-TEM to the Midtdalsbreen dataset

### 5.1 1D Resistivity Profiles

Using the data collected at Midtdalsbreen, we produced 1D resistivity profiles for the soundings acquired at the midpoint of lines A, B and C (Fig. 7). Anomalous data points either induced by residual current still in the transmitter (at early timegates) or below the background noise level (at later timegates) were removed, shown as "X" in Fig. 7(ii). Inversions were run with depth constraints taken from the GPR dataset for the snow-ice and glacier bed interfaces (red and blue horizons, respectively, in Fig.7). The same inversion parameters were used as the synthetic study (Table A1), but with the maximum depth extended to 160 m to test the limit of DOI.

Posterior resistivity distributions are shown in Fig. 7(i), comparison of data fit with the ensemble models are shown in Fig. 7(ii) and posterior distribution of the number of nuclei are shown in Fig. 7(iii). The estimated uncertainty for the most likely solution is calculated as one-half of the interquartile range at each depth, used due to non-normal pdfs. As is clear from Fig. 7(i), conductive layers identified within the subglacial material are well resolved with a tight pdf and low uncertainty. However, resistive layers (e.g., ice) have a wide pdf with large uncertainty estimates. For example, the uncertainty of the resistive ice layer is typically estimated as $\sim \pm 10^4$ $\Omega$m, whereas that of the conductive subglacial material is $\sim \pm 10^2$ $\Omega$m. The maximum DOI can also be identified in these distributions, expressed where the posterior distribution extends across the prior resistivity boundaries applied in a given layer (Table 2). This is 90 m for Line A, 87 m Line B and 76 m for Line C.

The 1D inversions show a ~10 m-thick layer, of $10^2$ $\Omega$m resistivity, directly under the ice, underlain in turn by conductive material of $10^0$-$10^1$ $\Omega$m for Line B and C. In contrast, Line A (off the ice margin, see Fig. 1b) shows a ~70 m-thick resistive layer, $10^4$-$10^5$ $\Omega$m, immediately beneath the snow layer, underlain by either a very conductive thin layer (~1 $\Omega$m) or a conductive material that extends to greater depth (these cannot be distinguished as it is close to the limit of DOI). We ensured that our methodology can discriminate the thick conductive layers observed in Lines B and C, and thin conductive layers underlain by resistive material approaching the DOI, using forward modelling. When compared (Fig.A4) to the data at the midpoint of Line C, the thicker conductive model has the closest resemblance to the observed data. This analysis suggests a spatially varying pattern of subglacial resistivities from the front of the glacier up to Line C.





**Figure 7. Results of the 1D soundings acquired at the midpoint of lines A, B and C inverted using MuLTI-TEM. i) resistivity distribution posterior probability distributions, with the maximum depth of investigation (DOI) plotted as the black dotted line. The blue and red solid lines highlight the snow-ice and ice-material depths.   ii) comparison of the observed data and 200 randomly chosen forward models from the model ensemble. The black X's show anomalous data points removed. iii) posterior distribution of number of nuclei.**

## 5.2 2D Resistivity Profiles

MuLTI-TEM is used to invert multiple independent 1D soundings acquired along Lines A, B, C (4 m intervals) and D (8 m intervals). Again, anomalous data points either induced by residual current in the transmitter (at early timegates) or below the background noise level (at later timegates) were removed, shown as the grey regions in Fig. 8 (left column). The raw signal acquired is generally above the background noise level for all time gates, except some anomalous points in the centre of Line A, corresponding with anomalous points at the NE end of Line D, highlighted in Fig. 8 left column.





When using a central loop TEM survey configuration the 1D response is simply located at the centre of the transmitter, where the receiver is located. However, with an offset transmitter-receiver survey configuration, used in this study, the location of the 1D sounding is a subject of debate. Some place the location below the receiver and others midway between the transmitter and receiver (Hoekstra and Blohm, 1990). The entire section between the transmitter and receiver is expected to

5   influence the measurements, especially at late times as the current is diffuse. In what follows, we assume the 1D location of each sounding to be at the centre point between the transmitter and receiver, although we note subsurface conditions near the transmitter may have a slightly larger influence on the received voltage measure at early times, when the current loop radius is approximately the same as the transmitter loop radius and not overlapping the receiver offset position.

Inversions were run with depth constraints supplied from snow and ice horizons picked from the GPR data and using

10   the same parameters as the synthetic study. We verified convergence of the solutions by running another Markov-chain and increasing the chain length to 1.5 million iterations: all tests reproduced the same posterior distribution. Consistent with initial observations in the 1D analysis, the 2D resistivity profiles (Fig. 8, central column) highlight a wide range of subglacial resistivity values, from $10^0$ to $10^5$ $\Omega$m, both along- and cross- glacier profiles, however there are some key consistent observations between these profiles, including

- a ~ $10^2$ $\Omega$m layer directly below the ice, for ice thicknesses < 20 m (mainly observed in Line B), which varies in thickness,
- a high resistive layer, $10^4$-$10^5$ $\Omega$m, in line A which matches line D at their intersection, and
- an lowermost layer of highly conductive material, ~$10^0$-$10^1$ $\Omega$m, that generally extends to the DOI

The estimated uncertainties for the mode resistivity solutions are displayed in Fig. 8 (rightmost column). This

20   reiterates previous observations made from the synthetic study and 1D analysis, that conductive layers identified within the subglacial material are well resolved by the TEM method with low uncertainty, however TEM methods struggle to fully resolve the more resistive layers which therefore have a larger uncertainty.





**Figure 8. 2D inversion outputs for Lines A-D from multiple 1D MuLTI-TEM inversions. Left column: received voltages input to MuLTI-TEM; central column: most likely 2D resistivity profiles; right column: estimated uncertainty (half the interquartile range of the posterior distribution). Snow and ice horizons are plotted in blue and red respectively.**





**6 Interpretation and Discussion**

**6.1 Joint interpretation of MuLTI-TEM with MuLTI seismic results**

The variability of resistivity ($10^0 – 10^5$ Ωm) in our TEM profiles suggests a complex subsurface structure, in which subglacial water may be in liquid and frozen states. However, the nature of the matrix – whether sediment or bedrock – cannot be

determined from resistivity alone. Resistivity is related to the resistivity of the pore fluids divided by the fractional porosity. A commonly used approximation is given by Archie's law, which states that:

$$R = aR_W/\emptyset^m \tag{5}$$

where $R$ is the bulk resistivity of a saturated porous medium, $\emptyset$ is the porosity, $R_W$ is the pore fluid resistivity and $m$ and $a$ are empirical quantities determined by the geometry of the pores (Archie, 1952). However, it is difficult to distinguish material

type, such as sediment or bedrock, from resistivity alone. By contrast, seismic shear wave methods are sensitive to the shear modulus, or stiffness, of a material but are insensitive to water content. However, a combined interpretation of resistivity and Vs profiles can be used to define a mutually-consistent system to characterise the material properties and water content of the subglacial environment. We do this using complementary seismic data, acquired alongside our TEM acquisitions in 2018.

The initial interpretation of the seismic data was presented in Killingbeck et al. (2019). This study excluded certain

phase velocities on the grounds that they were too high (Fig.A6); but this merit re-evaluation when compared with the co-located observation of high resistivity. Therefore, in this integrated interpretation, the high phase velocities are include, thus providing broader bandwidth dispersion curves.

Observing trends of Vs and resistivity in our profiles, three clear patterns emerge within the subglacial material:

    i)        zones of low Vs and low resistivity,

ii)       high Vs and high resistivity, and

    iii)      high Vs with low resistivity (Fig. 9; leftmost and centre columns).

These patterns have been used to define 3 different material types (Table 3). From previous electromagnetic and seismic studies (e.g., King et al., 1988; Schneider et al., 2013; Wu at al., 2017), liquid water in the pores of unconsolidated material has a low resistivity and low Vs (we define this as partially frozen sediment), whereas frozen water in pores is very resistive with a high

Vs (defined as frozen sediment/permafrost). We assume the bedrock comprises of phyllite, crystalline granite and gneiss with a high Vs and high resistivity. However, the resistivity profiles show the bedrock to have a very low resistivity suggesting it could be highly fractured and weathered with saline water in the fractures. The presence of saline water can decrease the electrical resistivity by as much as 9 orders of magnitudes (Olhoeft, 1981).



**Table 3. Vs and resistivity ranges for subglacial material lithologies, used in analysis of both MASW and TEM. Material types have been defined from King et al., 1988; Mikucki et al., 2015 and Killingbeck et al., 2019.**

| Material | Vs range (m/s) | Resistivity range (Ωm) |
|---|---|---|
| Partial frozen sediment/till | Vs < 1600 | 50 < R < 500 |
| Frozen sediment/permafrost | Vs > 1900 | R > 500 |
| Weathered/fractured bedrock with saline water | Vs > 1900 | R < 50 |

The resistivity and Vs profiles are linearly interpolated such that they have mutually consistent sample intervals and
depth extents, and are thus directly comparable. Killingbeck et al. (2019) consider abrupt lateral variations in the Vs profiles
to be noise, and interpret the broader variation in lateral and vertical character as the representative velocity structure.
Therefore, since the lateral resolution of the Vs profiles (estimated from the length of the geophone spread; 30-40 m) is larger
than that of the resistivity profiles (estimated from the horizontal spacing between 1D soundings; 4-8 m), we apply lateral
smoothing to the Vs profiles during the joint analysis. After these steps, we obtain a smooth joint interpretation of the predicted
subglacial material shown in Fig. 9 (rightmost column) and Fig. 10.

The joint interpretation shows zones of mainly sediment and thick permafrost (> 40 m) at the front of the glacier,
along line A, matching observations at the corresponding intersection point with line D. Note the sparse, disconnected areas
identified as bedrock in Line A are regarded as a misallocation, potentially due to abrupt lateral variations in Vs which we
regard as noise. Unfrozen sediment occurs directly below the ice at line B, with a varying thickness of 30 m at the eastern end
to 5 – 10 m towards the western end. In contrast, a mixture of frozen and unfrozen sediment is observed directly below the ice
at line C, typically ~ 10 m thick. Underlying the ice and sediment layers in all lines is the conductive bedrock with its structure
shown clearly in the cross glacier line, D. This also highlights a thin zone of frozen sediment directly under the glacier tongue,
matching observations from the GPR data shown in Reinardy et al. (2019), suggesting there is a frozen tongue.





Figure 9. Joint interpretation of Vs and resistivity profiles for lines A-D. Left column: modal Vs solution. Central column: modal resistivity solution. Example of areas with high Vs and high R is shown in Line A, low Vs and low R is shown in Line B and high Vs with low R is shown in Line D. Right column: estimated subglacial material when applying Vs and resistivity conditions of table 3. Note the sparse, disconnected areas identified as bedrock in Line A are regarded as a miss-allocation.

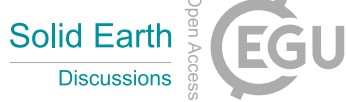

**Figure 10. i) 3-D cross-section of lines A-D, showing the subglacial material estimated from applying Vs and resistivity conditions stated in table 3. ii) Depth slice through 3-D cross at the top rock horizon. iii) Depth slice through 3-D cross-section at 38 m.**





## 6.2 Discussion

MuLTI-TEM combines a probabilistic approach with external depth constraints to mitigate ambiguous, non-unique solutions found in conventional TEM inversions. It provides a robust quantitative uncertainty analysis of any chosen model at all depth levels, also providing an accurate estimate of DOI using the posterior distribution. The addition of depth constraints improves

the characterisation of material, particularly beneath conductive layers and enables a faster convergence of the solution, with a better fit to the observed data (lower misfit), as demonstrated in the synthetic study.

Nevertheless, the success of MuLTI-TEM depends fundamentally on the input data quality and its suitability for the specific target imaged. With TEM methods, it is often not possible to determine separately the conductivity and thickness, only the conductance (product of thickness and conductivity) can be determined. Therefore, thorough synthetic modelling

should be undertaken before acquiring data in the field to determine if the survey design and time range of measurements is sufficient and suitable to detect specific targets.

Although this paper focuses on the specific TEM survey design used in this study, ground based 10 x 10 m transmitter with receiver 15 m away, MuLTI-TEM can be used with most TEM datasets. The Leroi forward modelling code can be used in frequency or time-domain mode to model most TEM transmitter/receiver combinations, ground based or airborne, and needs

only to be adapted to the user specific TEM configuration (Raiche, 2008). Equally, depth constraints are provided as numerical inputs and can therefore be supplied from any external source e.g., borehole measurements or any complementary inference from an external geophysical data source.

This paper has presented how a joint analysis of three geophysical datasets can increase our understanding of the material in the subsurface and provide a more detailed interpretation. Using GPR information as a depth constraint, we have

combined insight from TEM and seismic shear wave methods to provide a detailed characterisation of the material beneath the margins of Midtdalsbreen. Critically, TEM data reveal hydrological properties to which the seismic analysis was insensitive, whereas the seismic data indicate the varying stiffness of the subglacial material. Future extensions of this interpretative strategy would include a fully-coupled joint inversion strategy for resistivity and Vs, which could lead to a more accurate understanding of the subsurface structure (utilizing the structural similarities between resistivity and seismic velocity

(e.g., Wisén and Christiansen, 2005). Alternatively, petrophysical relationships could be derived to obtain and/or guide interpretations of the volumetric proportions of water, ice and air in the subsurface (e.g., Hauck et al., 2008). Such a combined approach would lead to more detailed analysis of the Midtdalsbreen margin, leading to a framework by which aquifer properties, such as porosity, water content and pore fluid conductivity/salinity, beneath large ice masses could be quantified. This could have a direct impact on basal parameters inputted to ice-flow models for a better prediction of ice motion over time,

and hence future sea level rise.



## 8 Conclusions

The material properties of the subglacial environment, in particular their water content and saturation, can be characterised by resistivity observations, obtained from TEM measurements. However, conventional TEM inversions provide solutions that are non-unique with no quantification of uncertainty estimates in depth and resistivity. This paper has presented the inversion algorithm 'MuLTI-TEM', used to overcome such problems. Our method uses a transdimensional Bayesian inversion approach adapted from the MuLTI algorithm (Killingbeck et al., 2018), which incorporates independent depth constraints to limit the solution space reducing ambiguity. Synthetic testing of multiple different scenarios representing a small glacier underlain by sediment showed the addition of depth constraints greatly improves numerical convergence and a reduction in misfit. This results in constrained solutions having a large improvement in the depth-average uncertainty of the output model, an average factor of 15 improvement on their unconstrained equivalents, with little computational power needed to obtain these results.

A joint interpretation, using Vs and resistivity boundaries, of the MuLTI-TEM results with MuLTI seismic surface wave results, presented in Killingbeck et al. (2019), considers three subglacial material classifications: sediment (Vs < 1600 m/s, 50 Ωm < R < 500 Ωm), permafrost (Vs > 1600 m/s, R > 500 Ωm) and weathered/fractured bedrock with saline water in the fractures (Vs > 1900 m/s, R < 50 Ωm). Their spatial extent, within the Midtdalsbreen's subglacial environment, shows a mixture of sediment and permafrost directly below the ice, and in the moraine at the front of the glacier, underlain by bedrock.

MuLTI-TEM is highly versatile being compatible with most TEM survey designs, ground based or airborne, as the Leroi forward modelling code can model most transmitter/receiver combination, along with the depth constraints being provided from any external source. This study presents novel methodologies, through MuLTI-TEM and MuLTI, by which other glacier and ice-sheets subglacial material can be explored, highlighting the importance of acquiring multiple geophysical datasets for accurately characterising the subglacial environment.

## 9 Code Availability

MuLTI-TEM can be found at: https://github.com/eespr/MuLTI-TEM, DOI 10.5281/zenodo.3351505.





## 10 Appendix

```
Offset Loop with 1 receiver - Model: 2 layers over basement, 2 platesCRLF
1 0 0 1 0                              ! TDFD, DO3D, ISYS, PRFL, ISTOPCRLF
0 4 20 1 1.05 1.05                     ! STEP, NSX, NCHNL, KRXW, REFTYM, OFFTIMECRLF
0.0      0.0CRLF
0.001    1.0CRLF
1.0492   1.0CRLF
1.05     0.0                           ! TXON, TXAMP(4)CRLF
0.006000  0.007625CRLF
0.007625  0.009750CRLF
0.009750  0.012500CRLF
0.012500  0.015880CRLF
0.015880  0.020250CRLF
0.020250  0.025880CRLF
0.025880  0.033000CRLF
0.033000  0.042130CRLF
0.042130  0.053750CRLF
0.053750  0.068500CRLF
0.068500  0.087380CRLF
0.087380  0.111400CRLF
0.111400  0.151700CRLF
0.151700  0.181100CRLF
0.181100  0.231000CRLF
0.231000  0.294600CRLF
0.294600  0.375900CRLF
0.375900  0.479500CRLF
0.479500  0.611600CRLF
0.611600  0.780100 →                   ! TOPN, TCLS (in ms)CRLF
1                                      ! SURVEY_TYPECRLF
1 1 1 1 4 1                            ! NLINES, MRXL, NTX, SOURCE_TYPE, MVRTX, NTRNCRLF
4 0                                    ! NVRTX TxZCRLF
5 5 0CRLF
-5 5 0CRLF
-5 -5 0CRLF
5 -5 0                                 ! SXE, SXN, SXZ(1,4)CRLF
1 1 1 1 4                              ! LINE IDTX, RX_TYPE, NRX, UNITSCRLF
3 0 0 1 0 1                            ! CMP SV_AZM, KNORM, IPLT, IDH, RXMNTCRLF
15 0 0                                 ! RXE, RXN, RXZ    CRLF
5 0 5                                  ! NLAYER, NPLATE, NLITHCRLF
  30 -1 1 1 0 0 1                       ! RES, SIG_T, RMU, REPS, CHRG, CTAU, CFREQ(1) - Lyr1CRLF
10000 -1 1 1 0 0 1                     ! RES, SIG_T, RMU, REPS, CHRG, CTAU, CFREQ(3) - Lyr2CRLF
100000 -1 1 1 0 0 1                    ! RES, SIG_T, RMU, REPS, CHRG, CTAU, CFREQ(6) - lyr3CRLF
1000 -1 1 1 0 0 1                      ! RES, SIG_T, RMU, REPS, CHRG, CTAU, CFREQ(6) - lyr4CRLF
1 -1 1 1 0 0 1                         ! RES, SIG_T, RMU, REPS, CHRG, CTAU, CFREQ(6) - lyr5CRLF
1 3                                    ! LITH, THICK - Layer 1CRLF
2 20                                   ! LITH, THICK - Layer 2CRLF
3 10                                   ! LITH, THICK - Layer 3CRLF
5                                    ! LITH, THICK - Layer 4CRLF
5                                      ! LITH, THICK - Layer 2CRLF
```

**Figure A1. Leroi input file specifically created to match the survey parameters used to acquire TEM data at Midtdalsbreen. The simplified Leroi forward modelling code used in MuLTI-TEM is created based on these parameters.**





**Table A1. Inversion parameters used in MuLTI-TEM for the synthetic feasibility study and 1D and 2D real data inversions, explained further in Killingbeck et al. (2018). Burn in number is the number of iterations discounted at the start of the chain to remove any dependencies of the initial conditions. Sigma resistivity, change, move and birth are user specified parameters that determine the magnitude of the four different perturbations that can be applied (change resistivity, move nucleus, give birth to a new nucleus, and remove a nucleus).**

| Inversion Parameter | Value |
|---|---|
| Number of Layers (non-constrained) | 1 |
| Number of Layers (constrained) | 3 |
| Weighting (data variance, σ) | 5% |
| Minimum number of total floating nuclei | 0 |
| Maximum number of total floating nuclei | 80 |
| Maximum depth | 80 m |
| Burn in number | 10 000 |
| Number of Iterations (including burn in) | 1 000 000 |
| Number of MCMC chains | 1 |
| Sigma resistivity change (log(R)) | 2 |
| Sigma move (meters) | 10 |
| Sigma birth (log(R)) | 2 |



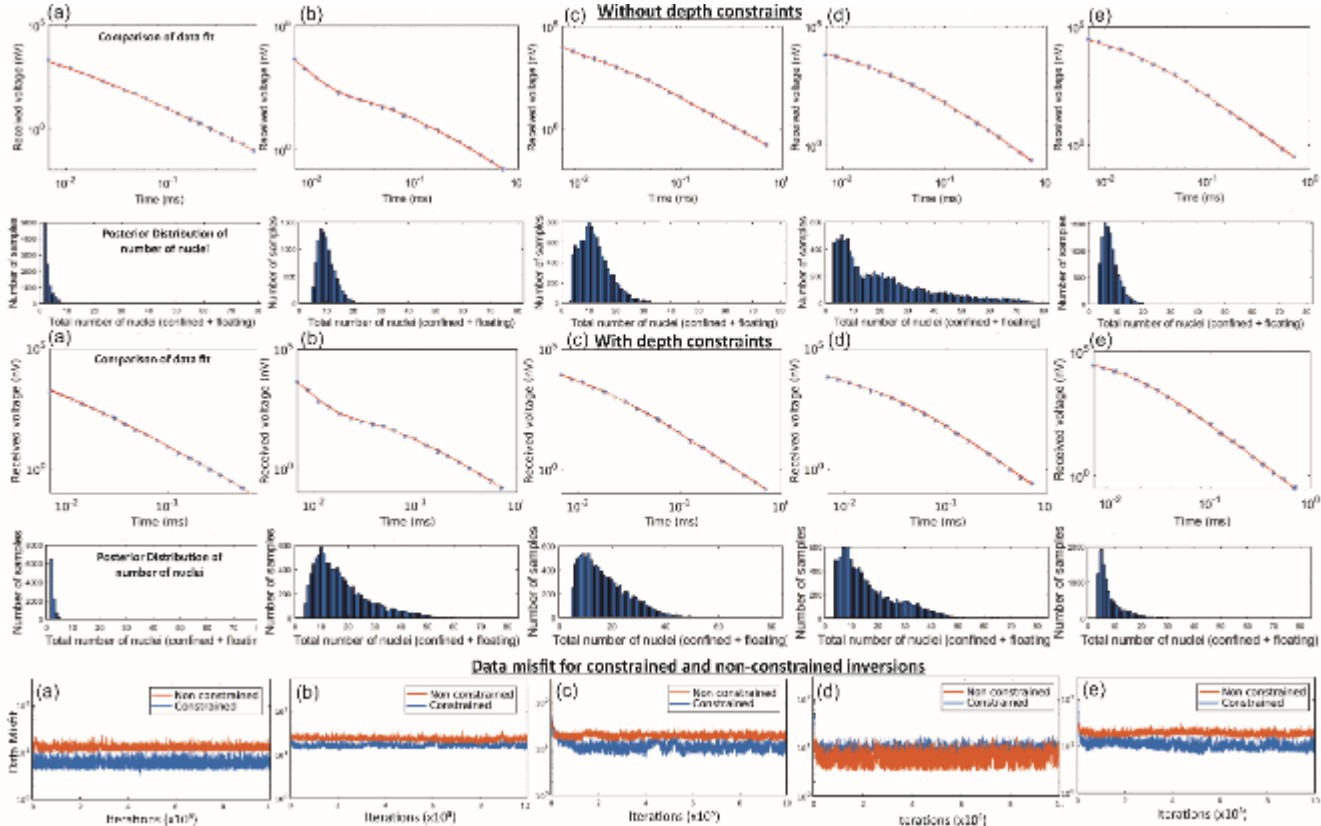

**Figure A2. Synthetic inversion results for all models in Fig. 4 (a-e), showing comparison of data fit and posterior distribution of number of nuclei plots.**

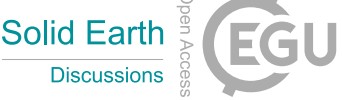

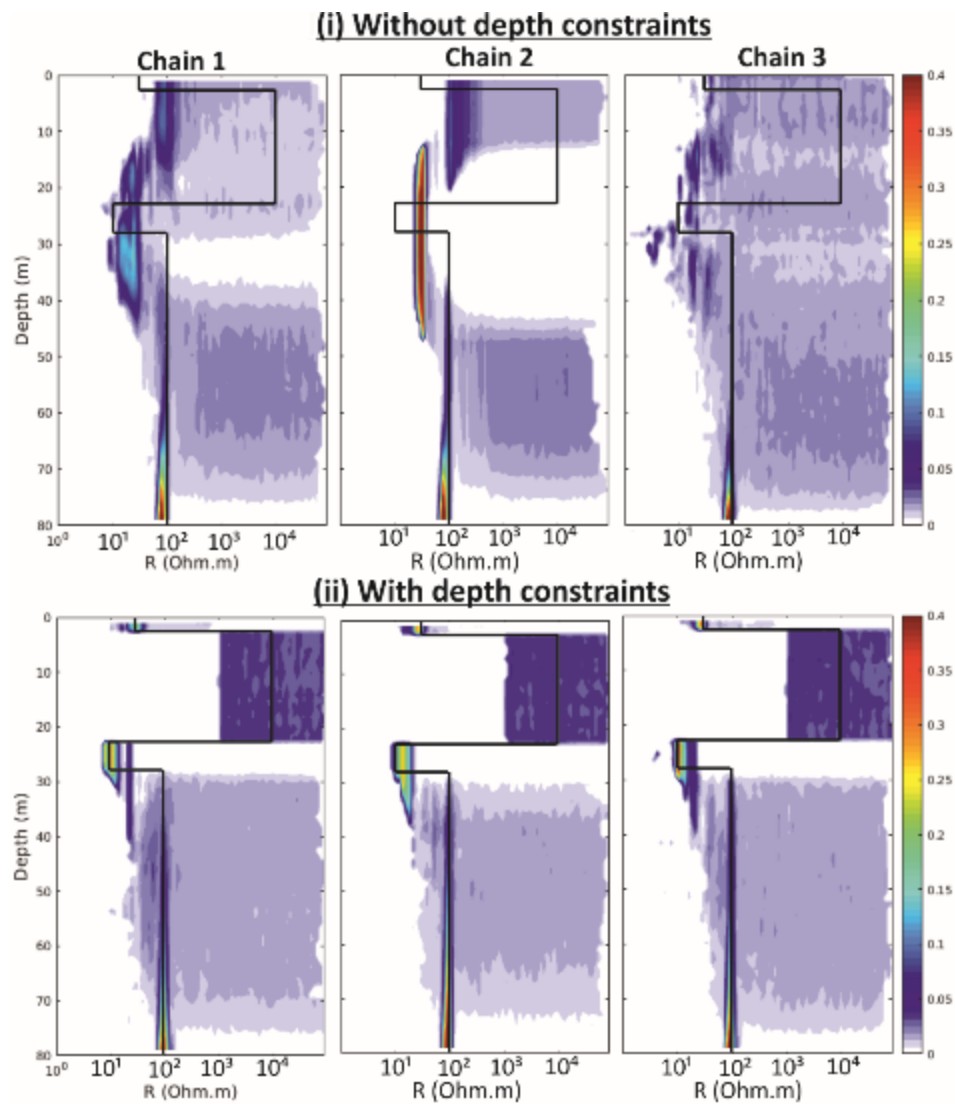

**Figure A3. Posterior distribution of resistivity for synthetic model d using 3 different chains for the non-constrained and constrained inversions with MuLTI-TEM, highlighting the independence of the constrained solutions on chain index, whereas the non-constrained distributions show a weak variation across chains.**



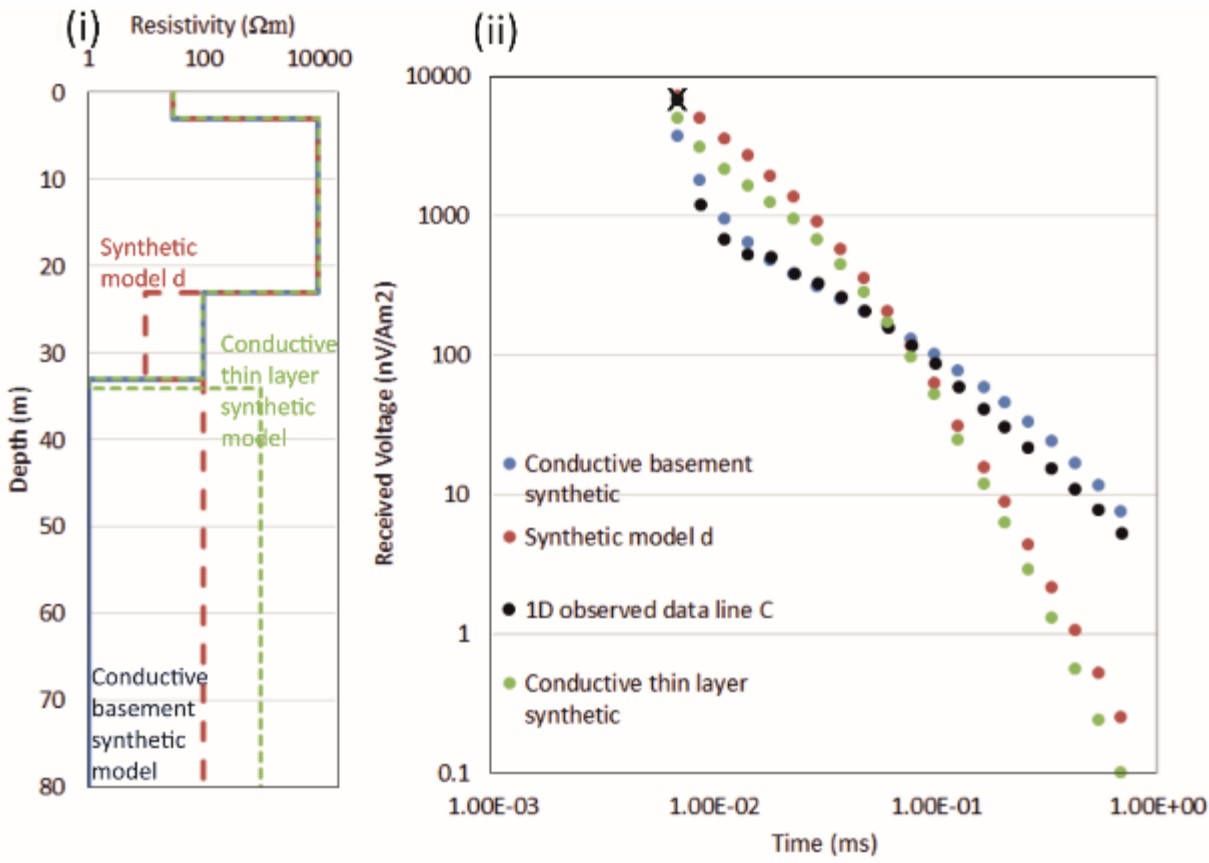

**Figure A4. i) Example synthetic models created with a conductive thin layer on top of a resistive basement (green) and a conductive basement (blue), compared to the original synthetic model d. ii) Calculated responses, using the Leroi forward modelling code, of the synthetics compared to the observed data at the midpoint of line C.**







**Figure A5. MuLTI-TEM inverted result for the a) conductive thin layer synthetic model and the b) conductive basement synthetic model, shown in Fig. A4.**





**Figure A6. i) Example of re-picked dispersion curves along line A at 20 m, 40 m and 70 m. The noisy high phase velocities observed > 55 Hz (originally not picked) are now thought to be real, matching the high resistivity observations along Line A. Red circles highlight the extra high frequencies picked. ii) Most likely resistivity profile with locations of dispersion curves marked by the black lines.**

## 11 Author Contributions

SK, AD, PL, LW designed the project. SK and AD acquired the data. SK and PL developed MuLTI TEM. RB provided advice and support while using the TEM equipment. SK prepared the manuscript with contributions from all co-authors.



## 12 Competing interests

The authors declare that they have no conflicts of interest.

## 13 Acknowledgements

This project was funded by the UK NERC SPHERES DTP, grant NE/L002574/1. Fieldwork was funded by the research project
'Snow Accumulation Patterns on Hardangerjøkulen Ice Cap (SNAP)', itself funded by the European Union's Horizon 2020 project INTERACT, under grant agreement No 730938. The time domain electromagnetic equipment was supplied by NERC Geophysical Equipment Facility, loan 1090. Fieldwork was greatly assisted by Emma Pearce, James Killingbeck and Kjell Magne Tangen. Alan Hobbs and all NERC GEF staff are thanked for their support and advice throughout the project.

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
