# Peer review of "Characterisation of subglacial water using a constrained transdimensional Bayesian Transient Electromagnetic Inversion"

_Solid Earth, 2019_

## Referee Comment (RC1) · Anonymous Referee #1 · 12 Sep 2019

General comments: This manuscript presents a good example of TEM survey to investigate the subglacial water. The trans-D Bayesian inversion is used to extract the resistivity information from the TEM data using the structural constraints to improve the accuracy. The seismic velocity is used to jointly delineate the material lithology. However, the manuscript is more likely to be a case history than a technical paper, as the trans-D Bayesian inversions is performed using the 1D forward operator from Leroi and the 1D trans-D Bayesian inversion has been reported in several literatures. On the contrary, the manuscript presents solid results in characterizing the subglacial water

jointly using the TEM, GPR and seismic data. Hence, I recommend the manuscript to be organized as a case history. Furthermore, the trans-D Bayesian inversion can be used to quantify the uncertainty of the inversion, which is its major benefit. However, the manuscript gives litter discussion about where the uncertainty comes from. Especially, the possible distribution that that the errors in the data obeys and how it is related to the chosen of the form of likelihood function. It would be good if this contains can be added.

Specific comments: 1. P2L13-14: "TEM methods . . .. . .by an offset transmitting coil". The grounded-wire or coincided loop are also commonly used to transmit the source signal of a TEM survey. 2. Compared with the methods mentioned in the INTRO-DUTION, Equation 1 only provides a poor estimates of DOI, without considering the current/noise level. Maybe the authors want to illustrate the role of transmitter loop size in controlling DOI, but it is well known that a larger loop will achieve a deeper earth. 3. Equation 2, 3,4. . . The vector and matrix should be bold. 4. P4L24-25: "since the accuracy of GPR depth estimation is $\sim$100-times smaller than the thinnest resolvable layer in TEM". Please add references for this statement. Another question: if the accuracy of the depth constraints from GPR results is higher, then what is the purpose of the variation of the dimensionality? 5. The elements of RJ-MCMC are outlined in the manuscript. It would be good if details of some key techniques are introduced, such as the principle to judge the convergence of the chain, how the accepted samples are resampled to suppress the correlations of the sample. 6. P11L4-5: "One million iterations were sufficient for the posterior distribution to converge (a test of 2 million iterations produced the same posterior)". Add the results of the test would be good. 7. P13L9-10: "data fit with the ensemble models are shown in Fig. 7(ii)". The model ensemble contains multiple models. Whose data fit is shown here, is it the average datafit? If it is, adding the error bar would be good. 8. P14 "5.2 2D Resistivity profiles". The 2D profiles is obtained by the stitching together the 1D inversion results or sampling the 2D model space as a whole. If the later one is chosen, how the 2D grids are updated? 9. Why the synthetic inversion of the 2D case is not given? 10.

Figure 10: As the presentation of the data is interpolated, please add the indicator of the recording station at the top of the figure so that the actual spacing of the stations can be illustrated.

Technical corrections: The manuscript is well written and I do not have technical corrections.

---

## Author Comment (AC1) · 28 Sep 2019

We thank the reviewer for their time reading our manuscript and their constructive comments and suggestions. We greatly appreciate all feedback provided. All comments are addressed as completely as possible, and we consider our manuscript much-improved as a result. The revised manuscript is attached as a pdf with all changes tracked. In our text below, the reviewers' comments are first, with our response directly beneath.

[Figure]

General comment: This manuscript presents a good example of TEM survey to investigate the subglacial water. The trans-D Bayesian inversion is used to extract the resistivity information from the TEM data using the structural constraints to improve the accuracy. The seismic velocity is used to jointly delineate the material lithology. However, the manuscript is more likely to be a case history than a technical paper, as the trans-D Bayesian inversions is performed using the 1D forward operator from Leroi and the 1D trans-D Bayesian inversion has been reported in several literatures. On the contrary, the manuscript presents solid results in characterizing the subglacial water jointly using the TEM, GPR and seismic data. Hence, I recommend the manuscript to be organized as a case history. Furthermore, the trans-D Bayesian inversion can be used to quantify the uncertainty of the inversion, which is its major benefit. However, the manuscript gives litter discussion about where the uncertainty comes from. Especially, the possible distribution that that the errors in the data obeys and how it is related to the chosen of the form of likelihood function. It would be good if this contains can be added.

Response: Thank you for your review and comments. We have already described the site in detail and presented a case history of Midtdalsbreen in Killingbeck et al., 2019 (referenced throughout this paper). Therefore, we did not organise this paper as a case history but rather an application of this method, MuLTI-TEM, of which the results have complimented and further improved our original case history presented in Killingbeck et al., 2019. We have added the sentence, and relevant references, in P5 L31 to address this further: "More detailed information on previous glaciological and geophysical studies on Midtdalsbreen can be found in Andersen and Sollid 1971; Etzelmüller and Hagen, 2005; Reinardy et al., 2013, 2019; Willis et al., 2012." However, we feel that our contribution carries more weight than a case-study, given that we develop existing approaches with the implementation of external depth constraints; indeed, we are grateful that the reviewer notes that our approach improves the robustness of our subsurface characterisation. We have emphasised this aspect of the work in the introduction to our paper, specifically in P2L7-11. With regards to the data uncertainty comment, we understand we have not been clear when discussing the uncertainty variable and where it comes from. We do say "estimate of their uncertainty ($\sigma$) derived from the variance of each data point calculated from the stack recordings". However, we have provided more details about this parameter in the manuscript by adding more detailed sentences in P5L14-15 stating it is dataset specific and defined for each different dataset, P9L11 clarifying we use 5% of the signal at each timegate, a similar noise model to Blatter et al. (2018) and P13L8-9 stating "The data variance ($\sigma$) was kept at 5% of the signal at each timegate as this was a good representation of the data variance of each data point calculated from the 3 stack recordings acquired in the field."

Specific comment 1: P2L13-14: "TEM methods . . .. . .by an offset transmitting coil". The grounded-wire or coincided loop are also commonly used to transmit the source signal of a TEM survey Response: Added grounded-wire and coincided loop to this sentence.

Specific comment 2: Compared with the methods mentioned in the INTRODUTION, Equation 1 only provides a poor estimates of DOI, without considering the current/noise level. Maybe the authors want to illustrate the role of transmitter loop size in controlling DOI, but it is well known that a larger loop will achieve a deeper earth. Response: We have altered the estimation provided in the introduction to that of Spies (1989), which explicitly takes account of the noise level.

Specific comment 3: Equation 2, 3,4. . . The vector and matrix should be bold. Response: This has been corrected.

Specific comment 4: P4L24-25: "since the accuracy of GPR depth estimation is âĹij100-times smaller than the thinnest resolvable layer in TEM". Please add references for this statement. Another question: if the accuracy of the depth constraints from GPR results is higher, then what is the purpose of the variation of the dimensionality? Response: We have changed this sentence to "In our case, in which constraints are drawn from high resolution GPR data, we consider depth constraints to be exact since the accuracy of GPR depth estimation is centimetre scale (Killingbeck et al., 2019), compared to the meter scale resolution of the TEM.". We have also added the appropriate reference which presents the methodology and results for the GPR depths and their uncertainty. The depth constraints provide information on the thickness of each layer enabling us to restrict the model space, by adding resistivity boundaries, of each specific layer. The variation in dimensionality allows resistivity variation within each defined layer, without having to fix the resistivity of that specific layer to one single value.

Specific comment 5: The elements of RJ-MCMC are outlined in the manuscript. It would be good if details of some key techniques are introduced, such as the principle to judge the convergence of the chain, how the accepted samples are resampled to suppress the correlations of the sample. Response: Convergence of the chain is performed simply by increasing the number of iterations and checking that the resulting distributions are the same. We have added Figure A2 in the Appendix to illustrate this. Furthermore, we "thin" the Markov chain (within the code this is controlled by the line "thin=100") to suppress correlations within the sampling chain and to speed convergence. This uses only every 100th model to compute the distribution statistics. We have added a sentence in the manuscript at P5 L25 to address this: "We thin the Markov chain by using every 100th model when computing the distribution statistics, which suppresses any localised correlations of neighbouring models and speeds up convergence."

Specific comment 6: P11L4-5: "One million iterations were sufficient for the posterior distribution to converge (a test of 2 million iterations produced the same posterior)". Add the results of the test would be good. Response: This has been added as a new figure, A2 in the appendix, and referenced in the above sentence within the main text.

Specific comment 7: P13L9-10: "data fit with the ensemble models are shown in Fig. 7(ii)". The model ensemble contains multiple models. Whose data fit is shown here, is it the average datafit? If it is, adding the error bar would be good. Response: This is a

comparison of the observed data and 200 randomly chosen forward models from the model ensemble. This is stated in the figure caption but not the main text, but we have added this to the main text to make it clearer.

Specific comment 8: P14 "5.2 2D Resistivity profiles". The 2D profiles is obtained by the stitching together the 1D inversion results or sampling the 2D model space as a whole. If the later one is chosen, how the 2D grids are updated? Response: The 2D profile is obtained by stitching together multiple 1D inversion results along the lines. This is already explained in the main text "...invert multiple independent 1D soundings acquired along Lines...".

Specific comment 9: Why the synthetic inversion of the 2D case is not given? Response: Because the 2D profile is obtained by stitching together multiple 1D inversion results (comment above), only synthetic inversions of the 1D case were needed.

Specific comment 10: Figure 10: As the presentation of the data is interpolated, please add the indicator of the recording station at the top of the figure so that the actual spacing of the stations can be illustrated. Response: We have added the location of the multiple 1D TEM soundings acquired along each line to Figure 10. The resistivity and seismic data were acquired in the same field season along the exact same lines, the acquisition parameters used in the field were chosen/set up so the two methods could be directly comparable with minimal interpolation needed. The active seismic acquisitions were performed with a Geometrics GEODE system and 48 10 Hz vertical-component geophones. For cross-glacier lines A, B and C, the source and geophone locations had 2 m intervals (where the CMP binning was 4 m); for the down-glacier line D, these were increased to 4 m (where the CMP binning was 8 m). These parameters are documented in Killingbeck et al., 2019. Multiple 1D soundings were recorded with the TEM system with 4 m intervals on the cross-glacier lines and 8 m interval on the down-glacier line, matching the seismic CMP binning for each line. However, the location of each 1D sounding and binned seismic CMP did not exactly match up, therefore, we interpolated and resampled both the resistivity and Vs solutions, originally sampled

every 4 m (Line A, B and C) and 8 m (Line D), to every meter so there was a resistivity and Vs data point at each meter along the line. This has been made clearer in the text by the addition of the sentences: "The resistivity and Vs profiles are linearly interpolated such that they have mutually consistent sample intervals (1 m) and depth extents (40 m). The TEM and seismic were acquired in the same field season along the same lines, the acquisition parameters were chosen so the two methods could be directly comparable. However, the location of each 1D TEM sounding and seismic common midpoint gather are offset by 2 m, therefore, we linearly interpolated and resampled both the resistivity and Vs solutions, originally sampled every 4 m (Line A, B and C) and 8 m (Line D), to every meter thus making them directly comparable."

Technical corrections: The manuscript is well written and I do not have technical corrections. Response: Thank you very much for your kind words.

Please also note the supplement to this comment:
https://www.solid-earth-discuss.net/se-2019-126/se-2019-126-AC1-supplement.pdf
* * *
[Figure]

**Supplement:**

[revised manuscript text omitted]

---

## Referee Comment (RC2) · Anandaroop Ray (Referee) · 7 Oct 2019

This is a generally well written paper by Killingbeck et al, which I recommend publication of on the condition that the following minor points be clarified:

Page 2 Line 8: It is better to expand TEM as Transient EM instead of Time Domain EM. If the authors wish to say Time Domain EM it is more appropriate to say TDEM.

Page 2 Line 13: "electromagnetic fields to investigate subsurface resistivity structure INDIRECTLY by measuring TRANSIENT eddy currents"

[Figure]

Page 2 Line 22: Please change eddy currents to "transient decay"

Page 3 Line 6: TEM / TDEM confusion

Page 3 Line 15: More time domain / transient confusion. For example, GPR can also be considered a time domain method, when analysis is done in the time domain. However, quasi-static diffusive EM geophysical methods, when analysed in the time domain, typically involve transients, hence the more appropriate "Transient EM" or TEM.

Page 3 Line 20: Change "time dependence" to "switch off"

Page 3 Line 21: The eddy currents PRODUCE a secondary EM field

Page 3 Line 22: The receiver TYPICALLY measures the induced . . . in the off periods. We can measure during on-periods too, it is harder to model.

Page 3 Line 25: The authors could mention somewhere around here that conductive material implies slower transient decay (e.g., Figure A4 of the manuscript) and sustenance of the induced subsurface eddy currents

Page 3 Line 3): Equation 1 is an approximation I believe, should be mentioned.

Page 4 Equations 2: t is not necessary in the data vector as time is not an observable. Just to be clear here, the authors should mention here that the mean recording in a stack window is used as data, and the variance of the mean (i.e., variance of the measurements divided by the number of measurements in the window) is the variance of the data. Population variance is not the variance of the mean. Also, the stack, through central limiting admits the use of a Gaussian likelihood.

Equations 3) We do not use the evidence constant for trans-D. Better to leave out p(d) and say p(m|d) \propto p(d|m) p(m)

Page 5 Line 8: Using depth dependent priors in a trans-D formulation is not strictly allowed (see Bodin and Sambridge 2009 for why this is so in the development of the prior). One can however use trans-D with depth independent priors and transform to

depth dependent values before modelling.

Page 11 Lines 20 onwards : As the authors point out on the following page, it is not surprising that better resolution is available when conductivity thickness tradeoffs are restricted. The authors could look at another approach of conditioning the posterior after inversion (Ray and Key 2012). However, I would recommend mentioning that the fixed interface depths may be allowed to vary, as the uncertainties on GPR interfaces are not as low as purported in the manuscript (see Ray et al 2016 for uncertainties on seismic reflectors and analogous wave physics, for example).

Also, the authors may try proposing from the prior for birth for better convergence (Dosso et al 2014).

Page 14 Section 5.2 and Figure 8: Marginal uncertainties along a 2D line can also be displayed instead of showing modal models, as shown by Ray et al 2014, Figure 11

In conclusion, since the authors have carried out probabilistic inversions of EM and shear wave dispersion data, I would recommend they try and present their conclusions on the facies classifications of the geology also in a probabilistic manner (or at least mention that this can be done). This requires some thought but will provide a much more informative set of displays than Figure 10.

All the best in your revisions, Anandaroop Ray

References

Bodin, T. & Sambridge, M. Seismic tomography with the reversible jump algorithm. Geophys. J. Int. 178, 1411–1436 (2009).

Dosso, S. E., Dettmer, J., Steininger, G. & Holland, C. W. Efficient trans-dimensional Bayesian inversion for geoacoustic profile estimation. Inverse Probl. 114018, (2014).

Ray, A. & Key, K. Bayesian inversion of marine CSEM data with a trans-dimensional self parametrizing algorithm. Geophys. J. Int. 191, 1135–1151 (2012).

Ray, A., Key, K., Bodin, T., Myer, D. & Constable, S. Bayesian inversion of marine CSEM data from the Scarborough gas field using a transdimensional 2-D parametrization. Geophys. J. Int. 199, 1847–1860 (2014).

Ray, A., Sekar, A., Hoversten, G. M. & Albertin, U. Frequency domain full waveform elastic inversion of marine seismic data from the Alba field using a Bayesian trans-dimensional algorithm. Geophys. J. Int. 205, 915–937 (2016).

---

## Author Comment (AC2) · 30 Oct 2019

We thank Anandaroop Ray for his time reading our manuscript and his constructive comments and suggestions. We greatly appreciate all feedback provided. All comments are addressed as completely as possible, and we consider our manuscript much-improved as a result. In our text below, the reviewers' comments are first, with our response directly beneath. The page and line numbers refer to the updated, edited, manuscript, which is attached as a pdf.

[Figure]

Specific comment 1: Page 2 Line 12: It is better to expand TEM as Transient EM instead of Time Domain EM. If the authors wish to say Time Domain EM it is more appropriate to say TDEM.

Response: We have changed "time-domain" to "transient".

Specific comment 2: Page 2 Line 18: "electromagnetic fields to investigate subsurface resistivity structure INDIRECTLY by measuring TRANSIENT eddy currents"

Response: We have updated this sentence with the added words above.

Specific comment 3: Page 2 Line 20: Please change eddy currents to "transient decay"

Response: This has been changed.

Specific comment 4: Page 3 Line 15: TEM / TDEM confusion

Response: Changed "time-domain" to "transient"

Specific comment 5: Page 3 Line 24: More time domain / transient confusion. For example, GPR can also be considered a time domain method, when analysis is done in the time domain. However, quasi-static diffusive EM geophysical methods, when analysed in the time domain, typically involve transients, hence the more appropriate "Transient EM" or TEM.

Response: Changed "time-domain" to "transient". We have also changed the title from "Time-Domain" to "Transient" to reflect this change.

Specific comment 6: Page 3 Line 29: Change "time dependence" to "switch off"

Response: This has been changed.

Specific comment 7: Page 3 Line 30: The eddy currents PRODUCE a secondary EM field

Response: This has been changed.

Specific comment 8: Page 3 Line 31: The receiver TYPICALLY measures the induced . . . in the off periods. We can measure during on-periods too, it is harder to model.

Response: "Typically" has been added.

Specific comment 9: Page 4 Line 3: The authors could mention somewhere around here that conductive material implies slower transient decay (e.g., Figure A4 of the manuscript) and sustenance of the induced subsurface eddy currents

Response: ". . . . . implying a slower transient decay" has been added to the end of this sentence.

Specific comment 10: Page 4 Line 6: Equation 1 is an approximation I believe, should be mentioned.

Response: "estimated" has been changed to "approximated"

Specific comment 11: Page 4 Line 22: Equations 2 t is not necessary in the data vector as time is not an observable. Just to be clear here, the authors should mention here that the mean recording in a stack window is used as data, and the variance of the mean (i.e., variance of the measurements divided by the number of measurements in the window) is the variance of the data. Population variance is not the variance of the mean. Also, the stack, through central limiting admits the use of a Gaussian likelihood.

Response: t has been removed from the data vector. The sentence before Equation 2 has been adapted to highlight the points above and make them clearer, Page 4 line 18: "The data input, d, to MuLTI-TEM are the voltages (v) at each of the N timegates (t), measured as the mean recording in a stack window. The mean, through central limiting, is assumed to be normally distributed with a variance $\sigma2$, the variance of the measurements divided by the number of measurements in the stack window, so that the data and uncertainties can be written as:"

Specific comment 12: Equations 3) We do not use the evidence constant for trans-D. Better to leave out p(d) and say p(m|d) \propto p(d|m) p(m)

Response: This has been changed.

Specific comment 13: Page 5 Line 14: Using depth dependent priors in a trans-D formulation is not strictly allowed (see Bodin and Sambridge 2009 for why this is so in the development of the prior). One can however use trans-D with depth independent priors and transform to depth dependent values before modelling.

Response: It is true that most authors assume that the prior subsurface structure is depth independent so that the prior analytically separates completely into its constituent parts (e.g. Bodin & Sambridge, 2009). However, as shown in Killingbeck et al. (2018), equation (7), in fact this is not necessary and the prior on the model (where the model includes both the depth and resistivity for each nucleus) can be written in terms of conditional probabilities. In this formulation, the prior is completely specified (but does not have a straight forward analytic form) and the usual trans-D methodology can be applied as usual. We have added the paragraph to page 5 line 14: "For the choice of prior distribution in transdimensional calculations, it is worth noting that usually the geophysical properties of the cells (here the resistivity) and the cell depths are assumed independent, allowing a simple separated analytic form for the prior distribution (e.g. Bodin and Sambridge, 2009). This is followed in our simplest geometry with no GPR constraints, for which the prior distribution on the resistivity is depth-independent and uniform with wide bounds on log(R) (e.g., R between 100-105 $\Omega$m), to convey the fact that no prior information (beyond that which can be reasonably assumed for typical materials) is known about the subsurface. However, by interpreting any GPR-derived layers as different materials (table 2) with much more narrowed ranges of resistivity, it is clear that a broad depth-independent prior distribution is no longer appropriate. Here we allow the prior distribution of resistivity to depend on depth, by defining for each layer a different uniform distribution that reflects the tightened bounds from lithological information. This restricted prior distribution then significantly decreases the number of permissible models describing resistivity with depth, reducing model ambiguity from any given set of data. In terms of the model parameters, the prior of the resistivity for

any nucleus is given by the specific layer that the nucleus is within (Killingbeck et al. (2018). Although a closed form expression for the depth-dependent prior distribution cannot be easily formulated, in the algorithm only the ratios of prior distributions are needed."

Specific comment 14: Page 12 Lines 20 onwards: As the authors point out on the following page, it is not surprising that better resolution is available when conductivity thickness tradeoffs are restricted. The authors could look at another approach of conditioning the posterior after inversion (Ray and Key 2012). However, I would recommend mentioning that the fixed interface depths may be allowed to vary, as the uncertainties on GPR interfaces are not as low as purported in the manuscript (see Ray et al 2016 for uncertainties on seismic reflectors and analogous wave physics, for example).

Also, the authors may try proposing from the prior for birth for better convergence (Dosso et al 2014).

Response: Thank you for these very interesting suggestions. In our case study, we are mainly interested in understanding the whole subglacial resistivity structure, which is completely unknown to us. Therefore, we do not have any specific hypothesis we wish to test at this stage. However, we understand the approach presented in Ray and Key (2012) could be a very useful analysis methodology for further studies. We have noted these suggestions in the manuscript's discussion section, Section 6.2. We have added the following sentence to page 22, line 6: "We note other methods could be used to enhance the efficiency of the transdimensional inversion, potentially providing better convergence rates, such as proposing the birth parameters from the prior (instead of a Gaussian distribution) e.g., Dosso et al., 2014. Further still, having access to the full posterior distribution enables subsets of the posterior model probabilities to be selected, testing various hypothesis about the model structure (Ray and Key, 2012)."

With regards to the fixed interface depths in MuLTI-TEM, this is a very good point we omitted to mention in our discussion, apologies. In our specific case study, the uncertainty in the constrained depths (derived from GPR) are very low compared to the uncertainty associated with the TEM inversion, hence why we fixed the depth constraints (page 5 line 2). However, we understand this is not always the case as pointed out in Ray et al., 2016. Therefore, we are currently developing the algorithm to be able to input the mean and standard deviation depths of each layer. We are working on a methodology where the layer depths are randomly perturbed (creating an extra perturbation step) within a normal distribution of the inputted mean and standard deviation, during numerical sampling of the posterior. We have added the sentence below to page 22, line 20: "In our Midtdalsbreen case study, the uncertainty in the depth constraints applied is negligible (deciimeter-accuracy from GPR data) compared to the observed data uncertainty (meter accuracy from TEM), motivating us to fix the internal interface depths. However, there remains a finite resolution in GPR data hence we are considering a modification to the MuLTI-TEM code to make it compatible with uncertain interface depths. This would also benefit depth constraints supplied from more uncertain data sources, thus making MuLTI-TEM more broadly applicable."

Specific comment 15: Page 15 Section 5.2 and Figure 8: Marginal uncertainties along a 2D line can also be displayed instead of showing modal models, as shown by Ray et al 2014, Figure 11.

Response: This is a really interesting point and the probability cubes displayed in Ray et al., 2014 are a very good visualization method for displaying and understanding the full solution, including its uncertainties, from the Bayesian inversion. As shown in Ray et al., 2014, this tool is very useful for characterising an anomalous target from a constant background resistivity e.g., a gas reservoir. Here, were are interested in the whole resistivity structure of the subglacial material, along all lines acquired, therefore we display the mode solutions. However, we have added the sentence below to the end of Section 5.2, page 16 line 22: "We note that marginal uncertainties along a 2D line can also be displayed as a 3D probability cube, with axes representing resistivity, line distance and depth, and colour bar representing the probability (e.g., Ray et al.,

2014). This aids visualisation of the Bayesian solution and its uncertainty, particularly useful when characterising an anomalous target from a constant background resistivity e.g., a subglacial aquifer or lake underlain by bedrock."

Concluding comment: In conclusion, since the authors have carried out probabilistic inversions of EM and shear wave dispersion data, I would recommend they try and present their conclusions on the facies classifications of the geology also in a probabilistic manner (or at least mention that this can be done). This requires some thought but will provide a much more informative set of displays than Figure 10.

Response: Thank you for your interesting suggestion and improvement for our concluding figure. It is possible to combine the facies classification in a probabilistic manner, however, we would need the joint probability distribution of Vs and R. If we assumed Vs and R were independent variables, we could combined the normalised pdf values, associated to the mode solutions for Vs and R, by calculating the product. However, realistically Vs and R are not independent variables, as they both depend on the same underlying subsurface and we do not have access to these conditional distributions, we only have the marginal distributions for each separate variable. Although, with a direct joint Vs-R Bayesian inversion we would be able to estimate this joint probability distribution and output the facies classifications in a probabilistic manner, this is currently being investigated as further work. We have adapted the following sentence in the manuscript to emphasis this point, Page 22, line 31: "Future extensions of this interpretative strategy could include petrophysical relationships to obtain and/or guide interpretations of the volumetric proportions of water, ice and air in the subsurface (e.g., Hauck et al., 2008). A further promising extension would be a modification to calculate the joint distribution of resistivity and Vs (rather than only the marginal distributions discussed in this paper) which could lead to a more accurate understanding of the subsurface structure (utilizing the structural similarities between resistivity and seismic velocity (e.g., Wisén and Christiansen, 2005). Such a combined approach would also lead resultto in more detailed analysis of the Midtdalsbreen

margin, including a probabilistic facies classification, leading to a framework by which aquifer properties, such as porosity, water content and pore fluid conductivity/salinity, beneath large ice masses could be quantified."

Please also note the supplement to this comment:
https://www.solid-earth-discuss.net/se-2019-126/se-2019-126-AC2-supplement.pdf

**Supplement:**

[revised manuscript text omitted]